# Children’s Relationships with a Non-Vertebrate Animal: The Case of a Giant African Land Snail (*Achatina fulica*) at School

**DOI:** 10.3390/ani13091575

**Published:** 2023-05-08

**Authors:** Katharina Hirschenhauser, Lisa Brodesser

**Affiliations:** Department of Science Education, University College for Education of Upper Austria, 4020 Linz, Austria

**Keywords:** animal assisted interventions, human animal relationships, animals at school

## Abstract

**Simple Summary:**

We explored the relationships of seven-years-old children with a Giant African land snail (*Achatina fulica*). The focus was on the potential effects of employing animals at school. One large snail named Bruno was kept inside a terrarium in a primary school. After seven months, the children’s relationship scores with Bruno were assessed and compared with the scores of same-aged children’s relationships with their vertebrate pets. The relationship scores with the snail were intermediate to high, comparable to the attachment of children to their dogs, cats, and rabbits. The results suggest that non-vertebrate species may have great potential for animal-assisted interventions in educational and therapeutic contexts.

**Abstract:**

Employing living animals in educational settings is popular and may assist learning. Human-animal relationships are considered fundamental for the effects of animal-assisted interventions (AAI) on successful learning. Key studies on AAI emphasize dogs, or other large-brained vertebrates, while AAI with non-vertebrate species is a yet rather unexplored field. However, bringing non-vertebrate species to school has ethical and practical advantages. In an exploratory study, we tested whether seven-years-old children would form caregiving relationships with a Giant African land snail (*Achatina fulica*). Prior to the survey, the snail had been kept inside a terrarium in the classroom for seven months. We employed a questionnaire for measuring children’s pet attachment to assess the children’s relationships with the snail. The observed relationship scores with the snail were intermediate to high and did not differ from same-aged children’s attachment scores with their dogs, cats, and rabbits. No differences due to gender were observed. Children potentially developed caregiving attitudes and empathy towards the snail, and thus, the presented results indicate potential benefits from employing a non-vertebrate species in educational settings, as well as for animal-assisted therapy. The specific features of *A. fulica* are discussed in the frame of human-animal interactions, learning, and anthropomorphism.

## 1. Introduction

Children are naturally attracted to animals, they are often interested and motivated to learn about an animal species or animals in general. Employing living animals at school provides an environment that enables joyful and successful learning, while boredom is reduced [1,2]. Moreover, animals at school provide occasions for social interactions and communication among the pupils [3] with beneficial effects at the social and the emotional level [4]. In sum, animals at school may enhance the environment for successful learning [5]. The underlying mechanisms of these effects are understood in the framework of adequate activation and regulation of stress during learning [6]. The human–animal relationship is considered a prerequisite for the effects of animal-assisted interventions (AAI) [7]. Evolutionarily, these effects take advantage of the biophilia phenomenon [8], which explains a natural attraction of living animals and nature in general to people. The biophilia hypothesis reasons that nature and natural environments are adaptive for humans and thus, the attachment of humans to animals seems to be an evolutionarily adaptive trait [9], which is probably a cross-cultural human universal [6].

The concept of attachment [6,10] is a framework for research on relationships between humans and pets. For children, a pet may even represent a secure attachment figure [4], comparable to the bonding with a parent or among siblings. Vice versa, the child itself may act as the caregiver for its pet, and the caregiving system is one basic element for the formation of attachment (including human–animal relationships). 

However, a relationship may also be regarded from the caregiver’s perspective, as by definition it “is a continuing and often committed association between two or more people, […] in which the participants have some degree of influence on each other’s thoughts, feelings, and actions.” [11]. The relationships of humans with many animal species, such as non-vertebrates, may be non-mutual. Such relationships are built on caregiving motivation and empathy at the side of the caregiving individual. In the current report, the focus is on the children as caregivers, and the caregiving relationship from the children towards the animal is regarded as a non-mutual phenomenon.

The literature on human–animal relationships considers the mutual nature of emotional responsiveness and social support a prerequisite of bonding and the development of attachment. Therefore, it has been considered relevant primarily with large-brained vertebrate species, in particular mammals and birds [6].

Key empirical studies on AAI in educational settings have been published on the beneficial effects of dog-assisted interventions at school (e.g., [5]). However, there are also limits and costs involved with dog training and responsibilities for owners during interventions and in daily private routines. Dogs are long-lived animals and relatively sensitive to stress, e.g., due to children’s behavior and noise in the classroom, and intensive care must be taken to avoid stress and to provide the well-being of the dog in the classroom. Some children may suffer from allergies (which is more prevalent with furry animal species), and some children may perceive disgust and aversion by the animal taken to school. Disgust is a subjective perception and may include dogs, as well as snails or insects [12,13,14]. Nevertheless, regarding time and resources, the employment of non-vertebrate species for AAI at schools may be advantageous as compared with school dogs. So far, the efficacy of employing non-vertebrate taxa for AAI in educational or therapeutic contexts is promising (e.g., sparking students’ interest and motivation, [15,16,17,18], but learning outcomes may be similar to employing film material [19]). However, studies on AAI with non-vertebrate taxa and, in particular, on the role of human–animal interactions are yet underrepresented in the literature.

We report an exploratory study on the relationships of seven-years-old children with a Giant African land snail (*Achatina fulica*) at school, i.e., an educational context. We were interested in whether children would potentially develop relationships also with a non-vertebrate animal—with the aim of understanding the potential of employing *A*. *fulica* for AAI. The potential relationships of children with the snail are understood in the context of caregiving attitudes and the development of empathy for the animal. The Giant African land snail is a moderately slimy mollusk of large body size and with a shell. It is an intriguing and silent animal, last but not least, due to its size—its shell may grow up to 27 cm [15]. Snails are easily kept, handled, and transported, as well as easily bred. If a warm (above 20 °C) and humid environment is maintained, the snails should remain active, but if conditions are temporarily allowed to become too dry or too cold, the snails are unlikely to suffer immediate, serious damage. They merely become inactive, retracting into their shells [16]. They are insensitive to noise, which is particularly relevant for interventions at school.

A large snail named Bruno was kept inside a terrarium in a classroom at a primary school. We anecdotally observed that children tend to anthropomorphize their experiences with the snail, for example, they said “Bruno enjoyed it today”, or “Bruno was curious”. Therefore, we hypothesized that the children may potentially develop a qualitative relationship with this non-vertebrate animal. After seven months, we assessed the children’s relationship scores with Bruno using a standardized questionnaire [7]. The aim was to compare the observed scores with the relationship scores of similar-aged children with their vertebrate pets from a previous study [20].

In the previous study, we assessed quantitatively the relationships of six- to ten-years-old children with their private pets by comparing the children’s individual attachment scores between pet species of different vertebrate classes [20]. In that study, we observed that the children’s relationships varied with the taxonomic order of their pet animal, i.e., they had higher relationship scores with dogs and cats than with guinea pigs, birds, or fish. It was assumed that the pattern followed the phylogenetic distance of the pet species to humans, which emerged almost in a “scala naturae” pattern. To test how the relationship scores of similarly-aged children with a snail would fit into this pattern, we employed the same questionnaire as in Hirschenhauser et al. [7,20] with the snail Bruno as the focal object of attachment.

## 2. Materials and Methods

### 2.1. Sample

In an Austrian primary school classroom with seven-years-old children (*n* = 15, 7 girls, 8 boys, mean age 7.1 ± 0.6 years), a Giant African land snail (*Achatina fulica*) was kept inside a terrarium for seven months. The children were involved in animal care and frequently interacted with the animal in an almost playful way during the breaks between lessons (Figure 1). The snail was named Bruno. Over the months, animal care had become a classroom routine, and the children were keen on providing different food options, observing preferences, exploring, and asking questions about the anatomy and the behavior of the snail.

### 2.2. Instrument

The children’s relationship scores with the snail Bruno were assessed using the same questionnaire [7] as in a previous study with vertebrate pets [20]. The questionnaire was specifically designed to measure attachment and relationship quality between adolescents and their pets (“My Pet and I”). Small adaptations were added for its use with primary school children (e.g., questions about trust, disgust, and caregiving attitudes). Beetz et al. [7] developed this questionnaire based on attachment theory for the assessment of the children’s internal working models rather than actual behavior, and for its applicability to a wide range of animals. The reliability of the questions is high for total score, trust, and communication (Cronbach’s alphas of 0.90, 0.85, and 0.87, respectively) and moderate for alienation (Cronbach’s alpha = 0.57) [21].

Twenty-one questions were answered using a 5-point Likert-scale, from 1 (“No, I disagree, this does not describe how I feel about Bruno”) to 5 (“Yes, I totally agree, this matches my feelings for Bruno”). The scale between 1 and 5 was provided as numbers and the meaning of the scale was explained to the children by the teacher. To assist them with reading, the questions were read to the children by the teacher. Following the original “Inventory of Parent and Peer Attachment” (IPPA, [22]) and Beetz et al. [7], the questions covered four subscales. These were about attachment in general, communication, trust, and alienation (Table 1). For the questions about the snail, adaptations were made in three cases (i.e., “I enjoy touching Bruno”; “I like feeding Bruno”, and “I feel disgusted by Bruno”). In total, eight questions were reversed scaled.

### 2.3. Analysis

The scores were treated as in Hirschenhauser et al. [20] to enable a direct comparison with the data for same-aged children’s relationships with their vertebrate pet [20]. Thus, the individual snail relationship scores were calculated by summing up the scores for selected questions within each subscale of the questionnaire. Following the method of Beetz et al. [7], three questions were selected within each of the three subscales communication, trust, and alienation (the latter was reversed scaled), and two questions from the dimension attachment (Table 1). Thus, the maximum relationship score to be reached was 43 (communication: 15, trust: 15, alienation: 3, attachment: 10). Relationship scores above 43 indicated answers in the context of high alienation (e.g., “It is useless to show my feelings to Bruno”).

Differences between the snail scores and those with other pet animals were tested using a Kruskal–Wallis test with pairwise DSCF comparisons. Gender effects were tested with a Mann–Whitney *U* test, age effects were tested employing a Spearman’s rank correlation between age and attachment scores. Statistical tests were conducted in Jamovi (version 2.2), and plots were created with GraphPad (Prism 5.01).

## 3. Results

The observed relationship scores of the children with the snail were intermediate to high, 41.6 ± 3.8 on average (with a range between 34 and 47), 80% of the children had individual scores ≥ 40. This range of relationship scores was similar to the pet-attachment scores of same-aged children with their dogs and other mammalian pets (Figure 2). Overall, there was a significant variance between all sampled animal taxa (Kruskal–Wallis test, *X*^2^ = 26.0; df 9; *p* = 0.002). The only significant pairwise difference was between dogs and fish (*W* = −4.47; *p* = 0.050). The pairwise test between the snail and fish was not significant (*W* = 1.28; *p* = 0.075). However, when the two single cases of one boy owning a mouse and another boy owning stick insects (Figure 2) were removed, the pet scores of both dogs and snails differed from fish (*X*^2^ = 23.0; df 7; *p* = 0.002; dog-fish: *W* = −4.47; *p* = 0.034; fish-snail: *W* = 4.28; *p* = 0.051). The variance of the observed scores was not due to gender (Mann–Whitney *U* test, *U* = 20.0; *n* = 15; *p* = 0.379) or age of the children (*r*_s_ = 0.1; *n* = 15; *p* = 0.705).

## 4. Discussion

We conclude from these first results that children potentially formed relationships with a Giant African snail, and thus, the known benefits of animal-assisted interventions (AAI) may also apply to a non-vertebrate animal species. In the current study, the children had seven months to observe the snail, experience its behavior and its preferences for food and shelter in the terrarium [23], and observe the anatomical specificities of the snail. Snails are exceptional animals to observe, as they are sensitive to tactile disturbance or rapid movements and respond immediately by retiring into their shell. The snail provided a chance for the children to experience patience during observations of the animal’s “silent activity” and thus, learning took place at a cognitive level, as well as beyond [14]. Behaviorally, there was a mutual human–animal interaction, which was probably catalyzed by the aforementioned children’s biophilic interest in the snail [8]. The children seemed to enjoy the experiences with Bruno, and in a playful way, they clearly had anthropomorphic interpretations of Bruno’s behavior. Anthropomorphism is another catalyst for the formation of bonds between humans and animals [24], and typically, children are particularly ready to do so.

Besides mutual interactions between children and the snail, we observed learning through experiences in the children and the unidirectional formation of relationships in the children towards the snail. Interactions were mutual, whereas it remains to be further tested whether learning may take place on both sides. Clearly, the children had learning experiences with the snail. However, it is unknown whether the snail has potential to be trained or to habituate to being touched or fed. There is only anecdotal evidence for individual behavioral differences in *Achatina fulica* (e.g., shy or exploratory individuals). Some snail keepers even suggested that their snail would be able to individually recognize human caregivers. So far, there is no evidence for individual recognition neither among *A. fulica*, nor between snails and humans. Thus, the observed relationships probably were a non-mutual and unidirectional phenomenon—the children were caregivers and developed empathy for the snail, however, obviously, not the other way around.

This non-mutual aspect sets a limit to the meaning of direct comparisons of the relationships between humans and their mammalian pet with the caregiving relationship of humans towards their snail (and not the other way around). Strictly taken, the observed caregiving relationships are not the same as the formation of attachment. Per definition, attachment characterizes “the different types of relationships between human infants and their caregivers […] which affect the infant’s later development and emotional stability.” [25]. Accordingly, “attachment” would focus on the snail’s perspective, and thus, the term is not applicable to the observed unidirectional attitudes of the children towards the animal. Therefore, the relationship scores should be interpreted carefully as indicators of the children’s activated caregiving system and development of empathy towards a non-vertebrate animal rather than attachment.

However, the observed relationships with the snail are not universally true for any non-vertebrate species. For example, Hirschenhauser et al. [20] reported the attachment score of one child owning Annam walking sticks (*Medauroidea extradentata*), and that score was well below the range of scores with the snail observed in the present study (14.0; Figure 2). However, Shipley and Bixler [13] observed a binary attitude of children towards insects and other non-vertebrate taxa between beautiful and bothersome. Walking sticks were not included in that study, however, snails were rated within the cluster for “fascinating” animals.

Snails provide some behavioral features that may support the development of a relationship. First, their immediate response to tactile disturbances (by retiring into the shell) allows learning about their preferences quickly. Second, these experiences are based on the concept of seeking proximity when the animal “likes something” and avoiding “what it doesn’t like”. This is a basic concept, which allows explaining several behaviors (in animals and humans), from simple responses to complex behavior patterns. Third, the potential for playful creativity during animal care supports anthropomorphic attributions, which engages mechanisms of social cognition and empathy [24].

The presented exploratory study has some methodological limitations. In addition to the small sample size, the descriptive data were assessed only once (after seven months of experience with the snail), and no repeated data are available. Thus, the observed relationship scores represent single measures, information on its sensitivity to changes is lacking, and thus, it cannot be located along a state-trait axis. However, even with the given limitations, the observed patterns provide the basis for further exploration of AAI with non-vertebrate animals, and in particular, the Giant African land snail.

## 5. Conclusions

Based on the presented results, we recommend exploring and learning more about the potential socio-emotional effects of employing non-vertebrate animal species for AAI in educational settings. This has ethical and practical advantages for animal-assisted interventions at school. The Giant African land snails, for example, are easily cared for and they are insensitive to noise. We encourage future studies to add more effect size to the presented observation, e.g., to test the physiological responses to skin contact between humans and snails and to learn more about the effects of experiences with selected species from other non-vertebrate taxa at school, such as insects or crustaceans.

## Figures and Tables

**Figure 1 animals-13-01575-f001:**
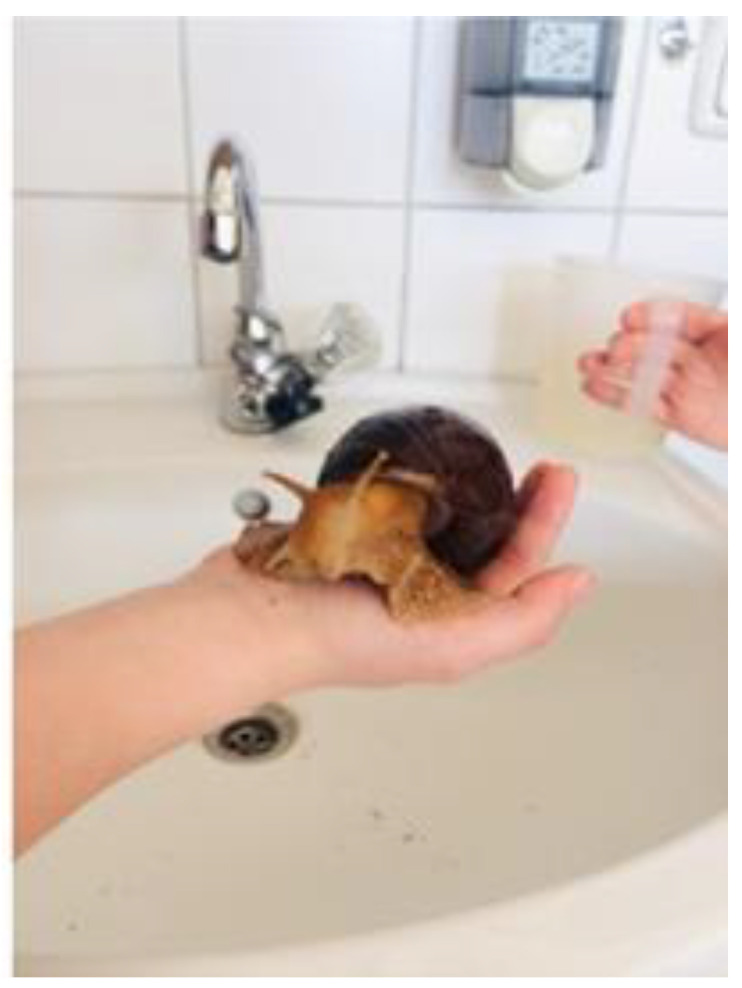
Children taking the Giant African land snail named “Bruno” for a shower in the classroom’s handbasin (photo © Lisa Brodesser).

**Figure 2 animals-13-01575-f002:**
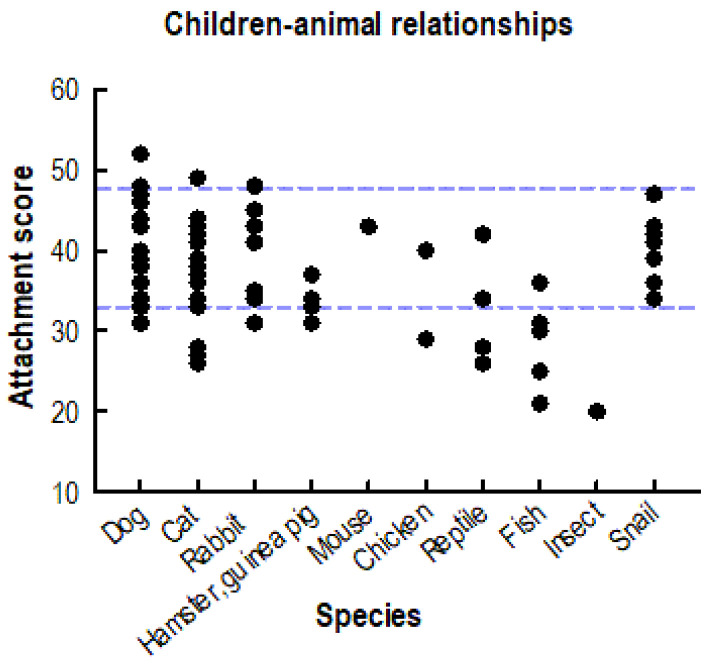
Children’s individual relationship scores with their vertebrate pets (modified from [20] and reprinted with permission of Taylor and Francis) and with the Giant African land snail (new data, added from the current study). The dotted blue lines indicate the range of scores with the snail.

**Table 1 animals-13-01575-t001:** The subscales and questions for calculating the individual attachment scores, following Beetz et al. [7] and Armsden and Greenberg [22].

**Dimension**	**Questions**	**Example**
Communication	4, 6, 21	“Sometimes I talk to Bruno about my feelings.”
Trust	1, 3, 18	“Bruno likes me as I am.”
Alienation	5, 8, 16	“Bruno does not pay much attention to me.”
Attachment	2, 13	“Bruno is a good friend.”

The questionnaire assesses attachment on a continuum between insecure (low trust and communication, high alienation) to secure (high trust, low alienation) [20].

## Data Availability

The data presented in this study are available on request from the corresponding author. The data are not publicly available due to privacy.

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
