# Peer review of "Children’s Relationships with a Non-Vertebrate Animal: The Case of a Giant African Land Snail (Achatina fulica) at School"

_animals, 2023, doi:10.3390/ani13091575_

Round 1

Reviewer 1 Report

The concept of attachment refers to interactions between children and their primary attachment figures. Children sent or display attachment signals especially when stressed or frightened (which is an expression of their activated attachment behavioral system). These signals activate the caregiving system of the child's caregiver which results in behavior suited to reduce the child's stress and anxiety. Thus a child's attachment experiences take place in the interplay between attachment and caregiving behavior as defined above. These experiences are internalized and finally form an internal working model of attachment. From this perspective it is implausible that the children in the current study made attachment experiences with the snail. A snail is not able to read a child's signals of stress and anxiety which is the prerequisite for adequate caregiving behavior (referred to as sensitivity in attachment theory). Without being sensitive to a child attachment signals the snail of course can not react adequately (referred to as responsively in attachment theory). I would assume that the snail is not able to display caregiving behavior at all. As far as I know snails even don't show caregiving behaviors towards their own offspring. Without a caregiving behavioral system a snail of course can't display caregiving behavior towards a child. From this perspective it is extremely unlikely that children for an attachment relationship with a snail. The authors should define attachment and caregiving more precise according to psychological standards, since these are psychological concept. Maybe the effects in this study can be explained because it was  the child who showed caregiving behavior. Maybe the child developed a caregiving relationship towards the snail which would be associated with similar effects.

Author Response

Reply to Review #1

We understand the reviewer’s important point(-s) and we appreciate the comments. In the revision we distinguished more clearly between mutual and non-mutual aspects of the phenomenon. We agree that the observed scores should be regarded as “caregiving relationship” if the relationship is defined as non-mutual – i.e. from child (as the caregiver) towards the snail (and not vice versa). The text has been revised accordingly, and the interpretation of the observed scales is more cautious now. We hope, it will satisfy this reviewer’s points.

We have realized, that in psychology the term “attachment” seems to be reserved for the child’s attachment towards its caregiver. Of course, we had never attempted to address the snail’s attachment towards its caregiver. The focus of this study was on the child as caregiver (and not the other way around) and the observed scores were, thus, understood as the children’s caregiving attitudes and empathy towards the snail. The previous version of the discussion dealt with this aspect as “bidirectional behaviour interactions”, whereas attachment was seen “unidirectional”. We replaced the terms by “mutual” and “non-mutual” and have rephrased parts of the text to address and explain this better.

In detail, we have added the psychological definitions of the terms “attachment” and “relationship” in the Introduction and the Discussion, and have rephrased the text accordingly. We have rephrased “attachment scores” to “relationship scores” throughout the text (including the title) to avoid misinterpretations.

The paragraph on the concept of attachment:
This part of the introduction has been extended to explain better the peculiarity of the snail case, and in the Discussion we have rephrased these parts, as well. (see also Reviewer #2)

Reviewer 2 Report

Literature review

Although there are some citations, they are not all considered with invertebrates and I suggest adding some more references that deal with invertebrates, or alternatively, reduce some of the references, such as about the Bald Ibis which does not seem relevant to understand the paper. Also, the references about dogs in the animal-assisted therapy could be reduced.

Examples could be https://www.mdpi.com/2075-4450/9/1/3

Line 45: although considered an important aspect of the introduction, despite being a short communication, I would like so to see more information about attachment, e.g., from general theory about attachment or attachment styles (at least in human interactions), but probably the attachment style theory has also been adopted to human animal interactions. I would be happy to see a fully fledged paragraph about attachment from a more psychological viewpoint, because readers of Animals may be less competent in psychology and need more explanation.

By the way, I full agree with your view about dogs – and I feel that AAI with dogs may not be beneficial, e.g. for the dogs (did somebody measure stress response in the dogs), as well as individual differences that are often neglected in the dog-companion-school literature. So it is good to see other taxa being involved in research.

The two-item attachment measures deserves some discussion because not all readers (including me) have the time to search in other publications how the scale was. Did you make EFA, CFA and validity issues, Cronbach’s whatever, readers would be happy to know more of it.

Methodologically I had some points: Was it a BACI measurement? Because comparing pre- and posttest scores would be very important? How long was the snail present? I feel that time could be an important factor for pre-post designs, and given the small sample size, a good alternative would be to collect more data during the process. Moreover, is the questionnaire you have used sensitive to changes? That said, can it be clearly located on the state-trait axis, as with anxiety measures or with PANAS? If and when attachment is a variable that is difficult to change (e.g., comparable to personality), a comparison is difficult, also, as it is not an experimental design, it also could be that many more factors in the long time span that the snail was present, may influenced the attachment scores (e.g., if and when attachment has some kind of developmental effect).

The figure shows data from another publication, I ould suggest that you make clearer in the legend that the snail data are new and added from the current study to avoid misunderstandings, e.g., people may assume duplicate publication, and for your own “safety” and to avoid this, it should be clearly labelled.

I have some concerns with the attachment scores of “birds” etc. because there are many birds (about 11.000 species), and attachment may vary from hummingbirds up to ostrichs and many studies in the “flagship topics” find extreme differences – also in pupils when comparing robin against an eagle etc., so you should have made clearer (probably also earlier) which species of birds, or at least order/family…

Given the small sample sizes, I would not expect gender differences or age correlations, this needs to be discussed in the discussion or even the tests could be left out – usually people make a power test when determining sample sizes based on previous studies, and in your case, it seems an exploratory study, so given the low sample size, statistical testing might be unnecessary.

Line 145 – it would be better to make a statistical test to see if the snail scores are significantly different from other taxa.

Discussion:

Line 157 – if you assume that children develop an attachment, this needs to be tested with a pre-post design?

The above noted concerns might be discussed in a limitations section.

Author Response

Reply to Review #2

We thank the reviewer for the appreciation and the very constructive comments. As far as possible, we have followed all suggestions.

  • Some more references that deal with invertebrates…:

    Thank you very much for the interesting link. We have added more references on non-vertebrates with focus on comparative aspects between different animal species, as well as the meta-analysis by Hummel & Randler (2012).
    The reference on the bald ibis study has been deleted. Reducing the references to dog studies was difficult because these cover some key results for the current understanding of AAI in school contexts, particularly with a focus on the human-animal-relationships and including papers reporting the use of the questionnaire (e.g. Beetz et al. 2011). To satisfy the requiry, the reference for Meints et al. (2022) has been deleted. However, there remain 5 references for studies dealing with school dogs, which we regard as important for the story line.

  • Introduction – more information about attachment theory, attachment from a mor psychological viewpoint:

    We have added more information on the issue in the Introduction and the Discussion, as this was also a major (and important) point raised by Reviewer #1. Consequently, we have had to realize that using the term “attachment” is not applicable (as it would address the attachment of the snail towards the child…). The observed scores probably address the children’s attitudes towards caregiving for the snail and their development of empathy towards the animal. Both aspects are key elements for the development of relationships and therefore, it is still of interest. However, we had to withdraw the term “attachment” for the snail case and replaced it by “caregiving relationships”.

  • Did somebody measure stress in the dogs:

    Yes, see for example Glenk et al. (2013) doi: 10.7120/09627286.22.3.369
    and Glenk et al. (2014) http://dx.doi.org/10.1016/j.jveb.2014.02.005
    and work from Adam Miklosi’s group

  • Did you make EFA, CFA and validity issues, Cronbach’s whatever…?

    The reliability of the questionnaire has been evaluated by Mayr (2007) using Cronbach’s alpha. We have added the information (as in the Anthrozoös paper) in the Methods.

  • Was it a BACI measurement?

    Unfortunately, we have no repeated data to compare scores between before and after the snail had been introduced to the children. However, as it is an instrument to assess relationship scores this would not have been applicable before introduction of the snail. As mentioned in Abstract, Introduction and Discussion, this was an exploratory and entirely descriptive study. (And therefore, we present it as a short communication rather than a full research paper). We have added this information in the newly added paragraph about the study’s limitations in the Discussion.

  • How long was the snail present?

    The snail was present in the classroom for seven months (has been mentioned repeatedly, in Abstract, Intro, Methods and Discussion).

  • Is the questionnaire sensitive to changes? Can it be located on the state-trait axis?

    As we present the descriptive data from one single assessment, we have no information on possible changes, or the the state-or-trait nature of the patterns. We agree that this would be very interesting and we have added it to the “limitations” paragraph.

  • Figure 2... make clear in legend that the snail data are new and added from the current study:

    Thank you for this comment! We have changed the figure legend accordingly.

  • Attachment scores of “birds”… make clearer which species/ family/ order of birds…:

    In this age class, there were only two children owning a bird pet – both had chicken. In the analysis in Hirschenhauser et al. (2017) this was termed “bird” to follow the argumentation line of a scala naturae pattern within the vertebrate taxa. However, we find this question very reasonable and were glad to add this information in the current paper - the label in Figure 2 has been changed to “chicken” accordingly.

  • Statistical tests for gender and age given the small sample size:

    As both tests may be employed with a sample size of 15, we kept this part. Of course, a larger sample size would be better. However, this study was restricted to one classroom and thus, sample size was restricted by the size of the class.

  • Add a statistical test to see if the snail scores are significantly different from other taxa:

    Thank you very much for this comment. We have run a Kruskal-Wallis test (as non-parametric equivalent to One-way ANOVA) and indeed, the snail sample is not different from the scores of mammalian pets. Overall, there are significant differences between all sampled animal taxa (X2=26.0; df 9; p=0.002). The only significant pairwise difference was between dogs and fish (W= -4.47; p = 0.050). The pairwise test between the snail and fish results in W= 1.28; p = 0.075. However, when the two single cases of one boy owning a mouse and another boy owning stick insects (Fig. 2) were removed, the pet scores of both dogs and snails differed from fish (X2 = 23.0; df 7; p = 0.002; dog-fish: W= -4.47; p = 0.034; fish-snail: W= 4.28; p = 0.051). These results have been added to the Results.

  • If you assume that children “develop an attachment”, this needs to be tested with a pre-post design?

    See above (BACI) – We agree that this would have been of interest. However, we assessed the relationship scores with the snail only once at the end of the seven months. Unfortunately, there were no repeated measurements. We have changed the first sentence of the Discussion accordingly (“… that children potentially form relationships with a Giant African land snail…”) and have added this comment to the paragraph on limitations in the Discussion.

Reviewer 3 Report

Thankyou for the opportunity to read what was a fascinating exploratory study.  The results of this study provide the basis for further exploration of AAI using non-vertebrates, and in particular the Giant African Land Snail.  The observations concerning the potential advantages of snails in classroom settings was particularly compelling. 

Informed Consent is recorded in the author statements at P6/L215.  However, there is no clear statement of the ethics review process for the study.  Furthermore, what if any considerations were given to safeguarding "Bruno" the snail.  It is important that such matters are addressed in practice and reported on in publication. 

For the administration of the questionnaire (P3/Ls110-125), please briefly explain how this was undertaken - did the children read the questions independently or were they read to them by a researcher; and were the scales only presented in words and numbers, or were they augmented with images / pictures? 

In the analysis (P3/Ls132-135), the reviewer might have missed something, but I was left wondering why the maximum score was 43:  11 items x max 5 points = 55 (as stated); but if 3 scores in the Alienation sub-scale were reversed scored this would still amount to a 15 point contribution to the total score?  Furthermore, if these scores were reversed in such a way that they were subsequently deleted from the total that would leave a total score of 45 (3 sub-scales x 5 points each = 15 points), and not 43 as stated?

In Table #1 (P4/Ls141-143), it might assist the reader to better understand the "dimensions if an additional column were to be added that provided a brief definition for each of - communication, trust, alienation, and attachment (these were not described in the introduction, and they are terms with particular psychological meanings that might not be well understood by the general readership of this journal). 

The results report analyses of the overall scores on "My Pet and I" (P4/Ls144-153).  Were any analyses undertaken at a sub-scale level (communication, trust, alienation, and attachment) and which might be available for reporting?

This paper sets an agenda of future studies.  It would have been good to have recorded interview data from the children (possibly structured around the same four domains as adopted in the analyses of the scored questionnaire.  Adopting a mixed-methods design in future studies could help to provide important explanations for the quantitative results.   

Author Response

Reply to Review #3:

Thank you very much for the appreciation and for sharing our interest in the subject. We have followed all suggestions except for adding detailed analyses within the subscales. Within the requested additional text parts the present manuscript provides some more explanations regarding the subscales within the limits of a “short communication”. We felt that the focus of this report should remain on the comparison with pet scores between same aged children and other animal species. Adding more subscale analyses would expand (at least) the Results chapter and distract from the focus of the study.

  • Add a statement of the ethics review process for the study

    We have added this information in the back end material.

  • For the administration of the questionnaire (P3/Ls110-125), please briefly explain how this was undertaken … and were the scales only presented in words and numbers, or were they augmented with images / pictures?

    Although the children in this study were able to read by themselves, the questions were read to the children by the teacher. We have added the information to the Methods-chapter 2.2:
    “The scale between 1 and 5 was provided as numbers and the meaning of the scale was explained to the children by the teacher. To assist them with reading, the questions were read to the children by the teacher.“

  • why the maximum score was 43…:

    The value 43 is correct. We have rephrased chapter 2.3 and hope, that the explanation is conceivable now:
    “Following the method of Beetz et al. [8], three questions were selected within each of the three subscales communication, trust, and *alienation (the latter was reversed scaled)*, and *two questions* from the dimension attachment (Table 1). Thus, the maximum relationship score to be reached was 43 (communication: 15; trust: 15; alienation: 3; attachment: 10). Relationship scores above 43 indicated answers in the context of high alienation (e. g. “It is useless to show my feelings to Bruno”).”

  • In Table #1 (P4/Ls141-143), it might assist the reader to better understand the "dimensions if an additional column were to be added that provided a brief definition for each of - communication, trust, alienation, and attachment

    We have added explanations on the subscales as a footnote to Table 1.
    “The questionnaire assesses attachment on a continuum between insecure (low trust and communication, high alienation) to secure (high trust, low alienation) [20]”
    We refrained from more details on this to keep this short report focused on the overall relationship score (Sum B) and its comparison with the scores in the previous study. Table 1 had been presented with the intention to present some detail on the nature of the questions – we would prefer to keep it small and simple. We hope, the footnote is useful and satisfying this request.

  • Were any analyses undertaken at a sub-scale level (communication, trust, alienation, and attachment) and which might be available for reporting?

    Please see above. In addition, some more details have been added to the Methods (see above) to explain the assessment of alienation.

Round 2

Reviewer 2 Report

thank you for responding to the comments